# Integrative multi-omics analysis uncovers tumor-immune-gut axis influencing immunotherapy outcomes in ovarian cancer

Spencer R. Rosario[1,2,14], Mark D. Long [1,14], Shanmuga Chilakapati[3], Eduardo Cortes Gomez [1], Sebastiano Battaglia[4], Prashant K. Singh[5], Jianmin Wang [1], Katy Wang[1], Kristopher Attwood[6], Suzanne M. Hess[7], AJ Robert McGray[7,8], Kunle Odunsi [9], Brahm H. Segal [10], Gyorgy Paragh [11], Song Liu[1], Jennifer A. Wargo [12,13] & Emese Zsiros [7] ✉

Recurrent ovarian cancer patients, especially those resistant to platinum, lack effective curative treatments. To address this, we conducted a phase 2 clinical trial (NCT02853318) combining pembrolizumab with bevacizumab, to increase T cell infiltration into the tumor, and oral cyclophosphamide, to reduce the number of regulatory T cells. The trial accrued 40 heavily pre-treated recurrent ovarian cancer patients. The primary endpoint, progression free survival, was extended to a median of 10.2 months. The secondary end-points demonstrated an objective response rate of 47.5%, and disease control in 30% of patients for over a year while maintaining a good quality of life. We performed comprehensive molecular, immune, microbiome, and metabolic profiling on samples of trial patients. Here, we show increased T and B cell clusters and distinct microbial patterns with amino acid and lipid metabolism are linked to exceptional clinical responses. This study suggests the immune milieu and host-microbiome can be leveraged to improve antitumor response in future immunotherapy trials.

Epithelial ovarian cancer, fallopian tube, and primary peritoneal cancer (collectively called OC) represent the deadliest gynecological malignancies[1]. Five-year survival with this disease remains less than 50% as most women ultimately succumb to treatment-refractory disease[2]. While the survival benefits of optimal cytoreduction and first-line platinum-taxane-based chemotherapy are well established, there

is a significant unmet need for effective, long-lasting treatments that also maintain a good quality of life (QoL) for patients with recurrent disease[3].

Approximately half of OC patients exhibit T cell infiltration into the tumor microenvironment (TME), a typically prognostic feature of increased survival[4,5]. However, even though up to 40% of OC tumors

[1]Department of Biostatistics and Bioinformatics, Roswell Park Comprehensive Cancer Center, Buffalo, NY 14263, USA. [2]Department of Pharmacology and Therapeutics, Roswell Park Comprehensive Cancer Center, Buffalo, NY 14263, USA. [3]New England Inflammation and Tissue Protection Institute, Northeastern University, Boston, MA 02111, USA. [4]Computational Biology Office of Translational Research, Janssen Pharmaceuticals, Buffalo, NY 14263, USA. [5]Department of Cancer Genetics and Genomics, Roswell Park Comprehensive Cancer Center, Buffalo, NY 14263, USA. [6]Department of Clinical Research, American College of Radiology, Reston, VA 20191, USA. [7]Department of Gynecologic Oncology, Roswell Park Comprehensive Cancer Center, Buffalo, NY 14263, USA. [8]Department of Immunology, Roswell Park Comprehensive Cancer Center, Buffalo, NY 14263, USA. [9]Department of Obstetrics and Gynecology, University of Chicago Comprehensive Cancer Center, Chicago, IL 60637, USA. [10]Department of Internal Medicine, Roswell Park Comprehensive Cancer Center, Buffalo, NY 14263, USA. [11]Department of Dermatology, Roswell Park Comprehensive Cancer Center, Buffalo, NY 14263, USA. [12]Department of Genomic Medicine, The University of Texas MD Anderson Cancer Center, Houston, TX 77030, USA. [13]Department of Surgical Oncology, The University of Texas MD Anderson Cancer Center, Houston, TX 77030, USA. [14]These authors contributed equally: Spencer R. Rosario, Mark D. Long. ✉e-mail: Emese.Zsiros@RoswellPark.org

express programmed cell death ligand-1 (PD-L1), only a small percentage (8–14%) of these patients respond to PD-1/PD-L1 immune checkpoint blockade (ICB)[6,7]. Moreover, these patients' median progression-free survival (PFS) is a mere 1.9–2.1 months. There are several factors contributing to the varied response to ICB in OC. For instance, high vascular endothelial growth factor (VEGF) expression can lead to irregular tumor vessels, hindering immune cell infiltration and function[8]. Furthermore, the absence of B cells and tertiary lymphoid structures (TLS) is linked to poor survival rates and diminished response to ICB[9,10]. Additionally, host factors, such as the composition of the fecal microbiome and the metabolites these microbes produce, can significantly impact antitumor immunity[11,12].

In our recently concluded single-arm phase 2 immunotherapy clinical trial (NCT02853318) in recurrent OC patients, we safely combined anti-PD-1 pembrolizumab with anti-VEGF bevacizumab (to prevent new tumor vessel growth and enhance T cell infiltration into the TME)[8] and oral cyclophosphamide (with the intent to deplete immunosuppressive regulatory T cells or $T_{regs}$)[13,14]. Our clinical trial showed a significant improvement in objective response rates (ORR) and PFS with this triple combination therapy, compared to the minimal responses in this patient population with ICB alone. Notably, 30% of the patients with a clinical benefit to the treatment achieved a durable response exceeding 12 months while maintaining an excellent QoL[14].

Here, we performed an integrative multi-omics analysis to understand the biological factors most strongly associated with durable clinical benefit. Using serially collected biospecimens, we investigated changes in the TME and evaluated gut microbial, metabolic, and immunological characteristics of patients with a durable clinical benefit (DCB) and those with limited clinical benefit (LCB). We identified unique transcriptional, immune, gut microbial, and metabolic signatures in DCB patients, highlighting the microbiome's role in shaping the antitumor immune response required for long-term disease control. These findings uncover several factors influencing therapeutic response to this effective ICB combination in OC patients and will help develop new strategies for future immunotherapy trials.

## Results

### Combining pembrolizumab, bevacizumab, and oral cyclophosphamide leads to improved patient outcomes

The NCT02853318 trial's treatment schema and the biospecimen collection time points can be seen in Fig. 1. The trial recruited 40 recurrent OC patients with a mean (SD) age of 62.2 (9.4) years. In addition, 30 participants (75.0%) had a platinum-resistant disease, with a mean (SD) number of prior lines of chemotherapy of 3.4 (2.4). No participants in this study had previous exposure to ICB as this treatment is only approved for patients with microsatellite instability (MSI-H), mismatch repair deficiency (dMMR), or high mutational burden (TMB-H)[15], which is only found in ~3% of the OC patient population[16].

This paper presents our trial's updated final clinical results, with an additional 3 years of survival data until November 2023 (Supplementary Fig. 1a–c). The median PFS on this trial was 10.2 months (primary endpoint), with an ORR of 47.5% (secondary endpoint), limited toxicity, and a good QoL measured by EORTC QoL questionnaires[14]. The combination therapy demonstrated clinical benefits (complete or partial response or stable disease) in 95.0% of patients. In addition to patients achieving a confirmed partial response or complete response as per iRECIST criteria in the clinical trial, individuals with PFS beyond the median of 10.2 months were classified as patients with DCB for the correlative analyses. This decision stems from the understanding that sustaining durable, stable disease in an immunotherapy trial signifies a considerable clinical benefit[17], especially for this cohort of patients, where the anticipated median PFS for second-line chemotherapy stands at 3–4 months and 1.9–2.1 months for pembrolizumab as a single agent[7,18,19]. These results represent a 2 to 3 times improvement compared to the average PFS typically observed in similar trials involving this patient demographic. This suggests that our approach provides a compelling new treatment strategy for recurrent OC and the NCCN Ovarian Expert Guideline Panel has now listed it on the 2024 NCCN guidelines as a second-line regimen in platinum-resistant OC[20].

Supplementary Fig. 1a displays a waterfall plot of the maximum reduction in tumor volume post-treatment, with patients divided into DCB and LCB groups. Each group's BRCA and PD-L1 status is also

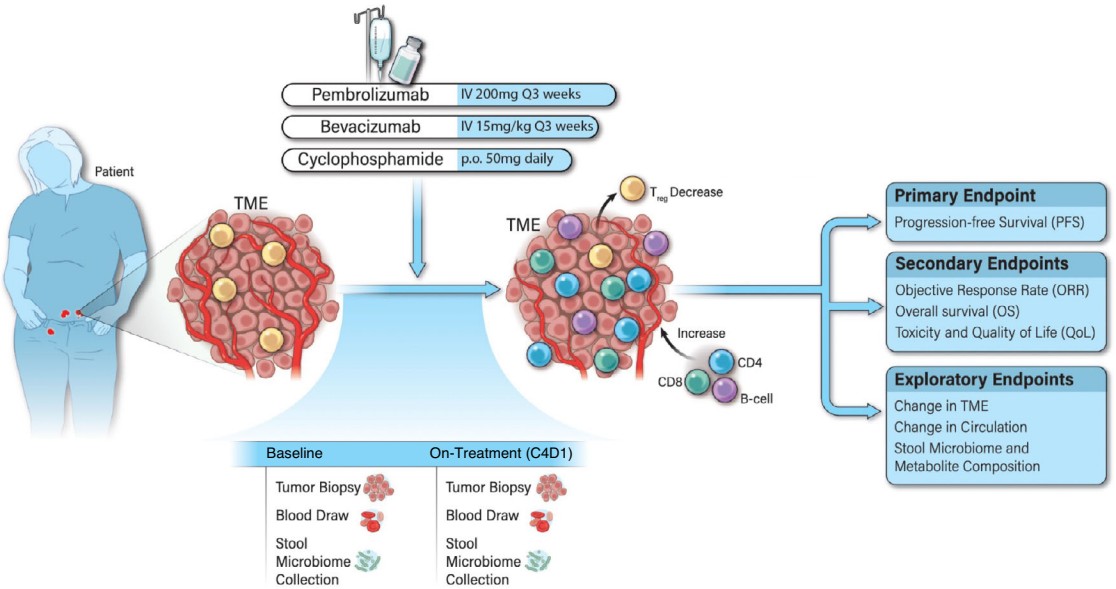

**Fig. 1 | Clinical cohort overview.** Recurrent ovarian cancer patients were previously enrolled in a single-arm phase 2 clinical trial combining pembrolizumab, bevacizumab, and oral metronomic cyclophosphamide therapy. Baseline (upon enrollment) and On-Treatment (Cycle 4, Day 1; C4D1) assessments, including tumor biopsies, blood samples, and stool collections, were obtained. The study diagram provides a thorough outline of the primary, secondary, and exploratory endpoints for the trial.

## Table 1 | Clinical characteristics of DCB and LCB patients

| Characteristic | NR, N = 19[a] | R, N = 21[a] | p-value[b] |
|---|---|---|---|
| Age | 64 (57, 71) | 62 (55, 66) | 0.2 |
| PDL1.status | | | 0.2 |
| Negative | 11 (58%) | 6 (29%) | |
| Positive | 7 (37%) | 12 (57%) | |
| Unknown | 1 (5.3%) | 3 (14%) | |
| BRCA.status | | | 0.5 |
| Negative | 12 (63%) | 11 (52%) | |
| Positive | 5 (26%) | 9 (43%) | |
| Unknown | 2 (11%) | 1 (4.8%) | |
| Platinum.senstivity | | | 0.069 |
| Resistant | 17 (89%) | 13 (62%) | |
| Sensitive | 2 (11%) | 8 (38%) | |
| TIL | | | 0.026 |
| – | 1 (5.3%) | 3 (14%) | |
| 1 | 4 (21%) | 7 (33%) | |
| 2 | 10 (53%) | 2 (9.5%) | |
| 3 | 4 (21%) | 9 (43%) | |
| Best response to treatment | | | 0.024 |
| irCR | 0 (0%) | 3 (14%) | |
| irPD | 2 (11%) | 0 (0%) | |
| irPR | 5 (26%) | 11 (52%) | |
| irSD | 12 (63%) | 7 (33%) | |
| Prior.BEV | 9 (47%) | 5 (24%) | 0.12 |
| Prior.Cycl | 4 (21%) | 1 (4.8%) | 0.2 |

Clinical characteristics of patient population.

Within TILs, – denotes not evaluable.

*irCR* immune related Complete Response, *irPD* immune related Progressive Disease, *irPR* immune related Partial Response, *irSD* immune related Stable Disease.

[a]Median (IQR); n (%).

[b]Wilcoxon rank sum exact test; Fisher's exact test; Pearson's Chi-squared test.

indicated. There was no significant correlation between BRCA variant status and PFS. However, BRCA-positive patients had a 71.4% ORR compared to 30.4% in BRCA-negative patients ($p = 0.02$). The expression of PD-L1 in baseline tumor biopsy samples did not correlate with ORR or PFS. Univariate analyses showed no significant differences in age, BRCA status, PD-L1 status, exposure to prior therapy, or initial sensitivity to platinum between the DCB and LCB groups (Table 1). However, significant associations were found with tumor-infiltrating lymphocytes (TILs) on immunohistochemistry (IHC), with more '3' scores observed in the DCB group.

Examining survival outcomes (Supplementary Fig. 1b, c) indicates that there are remarkable responses that extend beyond the initial 18-month period, with DCB still observed at 30 months (48% of DCB) and 40 months (29% of DCB) post-initial treatment. Notably, one DCB (5% of DCB) is still alive and nearing the five-year mark. In addition, the Kaplan–Meier analysis (Supplementary Fig. 1c) demonstrates a more than three-fold difference in median PFS between DCB and LCB, with DCB showing a PFS of 20.2 months compared to 5.72 months for LCB. These data confirm and expand upon the original clinical observations from the initial stop date[14], suggesting that some patients may experience benefits lasting up to five times longer than typical LCB.

Considering the notable differences in clinical outcomes observed between the two groups, we investigated further exploratory endpoints, such as TME changes and fecal microbiome or metabolite profile variations. Our findings indicate that differences in the tumor-immune-gut axis are associated with treatment outcomes and could be utilized to enhance treatment approaches in upcoming immunotherapy clinical studies.

## Exceptional response to ICB combination therapy associates with a highly functional and diverse immune TME

We conducted transcriptomic profiling (RNA-seq) on the serially collected baseline and on-therapy (On.TX, after 3 cycles of treatment on cycle 4 day 1) tumor biopsies from 29 patients with high quality samples on trial to characterize the molecular differences in tumors with divergent clinical responses to ICB combination therapy. We aimed to identify differentially expressed genes in DCB vs. LCB at baseline (Baseline_DCB vs. Baseline_LCB) and on treatment (On.TX_DCB vs. On.TX_LCB) (Fig. 2). Additionally, changes associated with therapy within DCB (On.TX_DCB vs. Baseline_DCB) and LCB (On.TX_LCB vs. Baseline_LCB) are shown in Supplementary Fig. 2a (Supplementary Data 1). Gene set enrichment analysis (GSEA) of tumor transcriptomes identified elevated T- and B cell activation, differentiation, and proliferation signatures in baseline DCB compared to LCB, further enhanced with treatment only in DCB (Fig. 2a, b). In addition, tumors from DCB patients were enriched for immune signatures associated with CD40, antigen presentation, cytokine production and signaling, and TLS[21,22], at baseline and On.TX. Conversely, MYC and neuronal signatures were primarily expressed in LCB tumors at baseline and On.TX.

Microenvironment cell population (MCP) counter[23] deconvolution scoring confirmed signatures of high immunogenic activity associated with DCB. Baseline and On.TX were associated with elevated immune population estimates, including CD8+ T, B, dendritic cells (DC), and macrophages/monocytes (Fig. 2c, d). Results from the MCP counter were highly concordant with estimates derived from 5 additional deconvolution scoring methods (Supplementary Fig. 2b). These findings suggest that patients who responded well to the combination ICB therapy had elevated and complex tumoral immune cell infiltration at baseline and after treatment compared to those who did not respond. To better understand the divergent phenotype of "immune hot" and "immune cold"[24] patients, we further assessed immune populations in the TME.

## Combination ICB induces stromal to tumoral immune trafficking in responding patients

Digital spatial profiling (DSP) of a select panel of 52 distinct tumor and immune-oncology markers was performed on serial biopsy material from 14 patients (DCB; n = 7 and LCB; n = 7). From each biopsy section, 3 regions of interest (ROI) were selected from both pathologically defined tumor and stromal compartments, totaling 168 ROI for analysis across patients. Representative imaging of ROI from patients at baseline and On.TX revealed the common presence of CD3+ aggregates in stromal regions at baseline and On.TX in both DCB and LCB patients, but these features were limited in tumoral areas at baseline indicating active lymphoid infiltration exclusion prior to treatment (Fig. 3a, Supplementary Fig. 3). Notably, CD3+ aggregates in tumoral regions were commonly observed On.TX in DCB suggests enhanced immune trafficking into tumoral areas associated with durable response to combination ICB.

Quantitative assessment of the complete set of immune markers revealed considerable immunogenic heterogeneity across ROI (Fig. 3b, Supplemental Fig. 3). Principal component analysis of ROI abundances revealed components primarily explained by response classification (DCB vs. LCB; PC5) (Supplementary Fig. 4a); however, a majority of variation across ROI was most readily defined by location (tumor vs. stroma; PC1) (Supplementary Fig. 4a). At baseline, tumor regions were distinguishable from the stromal areas by an abundance of ovarian epithelial markers, including epithelial cell adhesion molecule (EPCAM), Estrogen Receptor alpha (ERα), Progesterone Receptor (PR), human epidermal growth factor receptor 2 (HER2), and the proliferative marker Ki-67 (Fig. 3b, Supplementary Fig. 4b, c). Conversely, stromal regions were associated with a substantial abundance of lineage-defining immune markers (CD3, CD8, CD4, CD20, CD14,

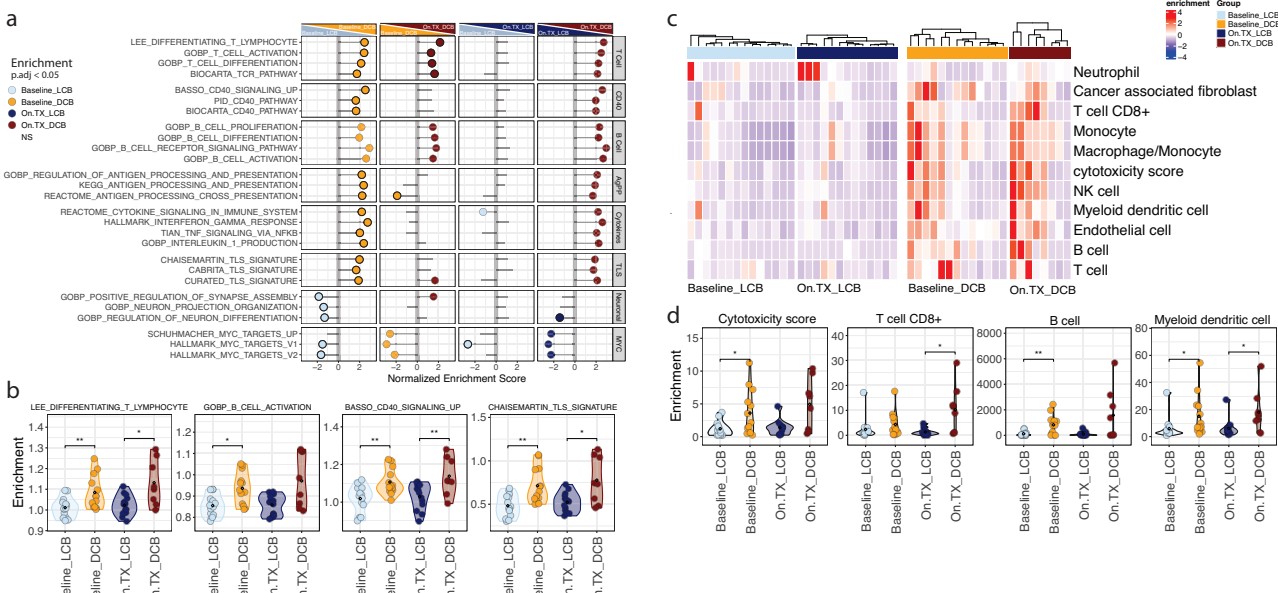

**Fig. 2 | Analysis of tumor transcriptomes indicates patients with durable clinical benfit exhibit greater immune activation compared to patients with limited clinical benefit. a** Through Gene Set Enrichment Analysis (GSEA) of tumor transcriptome data, immune population changes are found to be enriched in durable clinical benefit (DCB) groups, with this enrichment further amplified during treatment (On.TX) compared to limited clinical benefit (LCB) groups. Additionally, MYC-associated and neuronal signatures are more prevalent in LCB groups compared to DCB groups, irrespective of the timepoint (Baseline or On.TX). **b** Quantification of immune-specific signatures based on GSEA, demonstrates

increases in immune-associated transcripts in DCB, as compared to LCB populations, including those associated with T cells, B cells, CD40 signaling, and tertiary lymphoid structures (TLS). Further, this corresponds with (**c**) enrichment of MCP Counter immune deconvolution signatures in DCB patient populations compared to LCB patient populations at baseline, and further exacerbated by treatment. **d** This is further exemplified in comparing MCP counter cytotoxicity, T cell, B cell, and myeloid dendritic cell (DC) signatures. *$p < 0.05$, **$p < 0.01$, two-sided Mann–Whitney U-Test. Exact $p$-values can be found in Data Table 1.

CD11C), confirming a complex, but tumor-excluded immune environment prior to treatment in most patients regardless of response status (Fig. 3b).

Differential abundance analysis revealed no significant treatment-induced shifts in marker abundance or localization in tumor or stromal regions in LCB (Fig. 3c–e). Conversely, numerous immune markers were elevated with treatment in DCB, particularly in defined tumor regions. These included lineage-defining markers (CD3, CD11C, CD20), but also cytotoxic effector (GZMB), co-stimulatory (CD40, CD27), and immune checkpoint (PD-L1, VISTA) proteins. Conversely, Ki-67 was reduced in DCB with treatment. Comparison of On.TX DCB to LCB similarly revealed enhanced tumor and stromal compartments' immunogenicity, indicating that response was associated with treatment-induced immune cell infiltration of the TME and trafficking to tumoral regions. Across ROI, strong correlations were observed between CD20, CD3, and CD11C, indicating co-occurrence of T, B, and DC, respectively, supportive of TLS formation. Using these combined marker abundances to define a TLS score revealed significant enrichment with DCB at baseline and On,TX (Fig. 3e, Supplementary Fig. 4c, d), supporting observations of TLS score enrichment from transcriptome analysis. These results point to altered patterns of immune cell infiltration, trafficking, and activity in DCB and LCB at baseline and On.TX; however, the biological etiology underlying this divergence is largely unknown.

## Response to combination ICB and outcomes correspond with microbiome alterations

Given strong evidence for a role of gut bacteria and bacterial supplementation in mediating response to immunotherapy[25], we assessed microbiome differences between DCB and LCB. Serially collected fecal samples were analyzed via 16s-rRNA-sequencing at baseline and On.TX. The microbial compositions showed no significant alpha or

beta bacterial diversity differences among response groups (Supplementary Fig. 5). However, a differential analysis comparing operational taxonomic unit (OTU) abundances, converging at the species level, in baseline DCB to LCB revealed considerable differences in microbial community composition (Fig. 4a). In line with observations from tumor transcriptomics and DSP analyses, DCB patients were bacterially distinct from LCB at baseline and On.TX which indicates a specific microbiome composition may be associated with a combination ICB response. We found several species associated with LCB (Supplementary Fig. 6), like *Enterococcus faecalis*[26] and *Clostridium perfringens*[27], which have been previously associated with resistance to immunotherapy. Conversely, we identified several highly abundant species in DCB, like *Intestinimonas butyriciproducens* and *Anaerotignum propionicum (Clostridium propionicum)*[28], highly consistent with prior reports suggesting these butyrate-producing bacteria are associated with enhanced response to immunotherapy[25]. We also identified bacterial species like *Dielma fastidiosa*, a newly recognized genera of bacteria[29], which were also present at high abundance in DCB.

Furthermore, the presence of several species was altered upon exposure to therapy, like *Ruminiclostridium spp* being detected at higher levels On.TX in DCB as compared to LCB (Supplementary Fig. 6a; bottom left). Notably, several bacterial species remained consistently elevated in DCB compared to LCB at baseline and On.TX (Fig. 4b, Supplementary Fig. 6a; bottom right), like *Dielma fastidiosa*, *Ruminococcus spp*, and *Peptococcus spp*. Given that these species are consistently abundant at high levels throughout treatment in DCB (Fig. 4b), they may offer potential as prognostic indicators or points of therapeutic leverage to help sensitize patients to combination ICB using probiotic supplementation.

Further, many of these bacterial species with elevated abundances at baseline in DCB were, not surprisingly, associated with prolonged

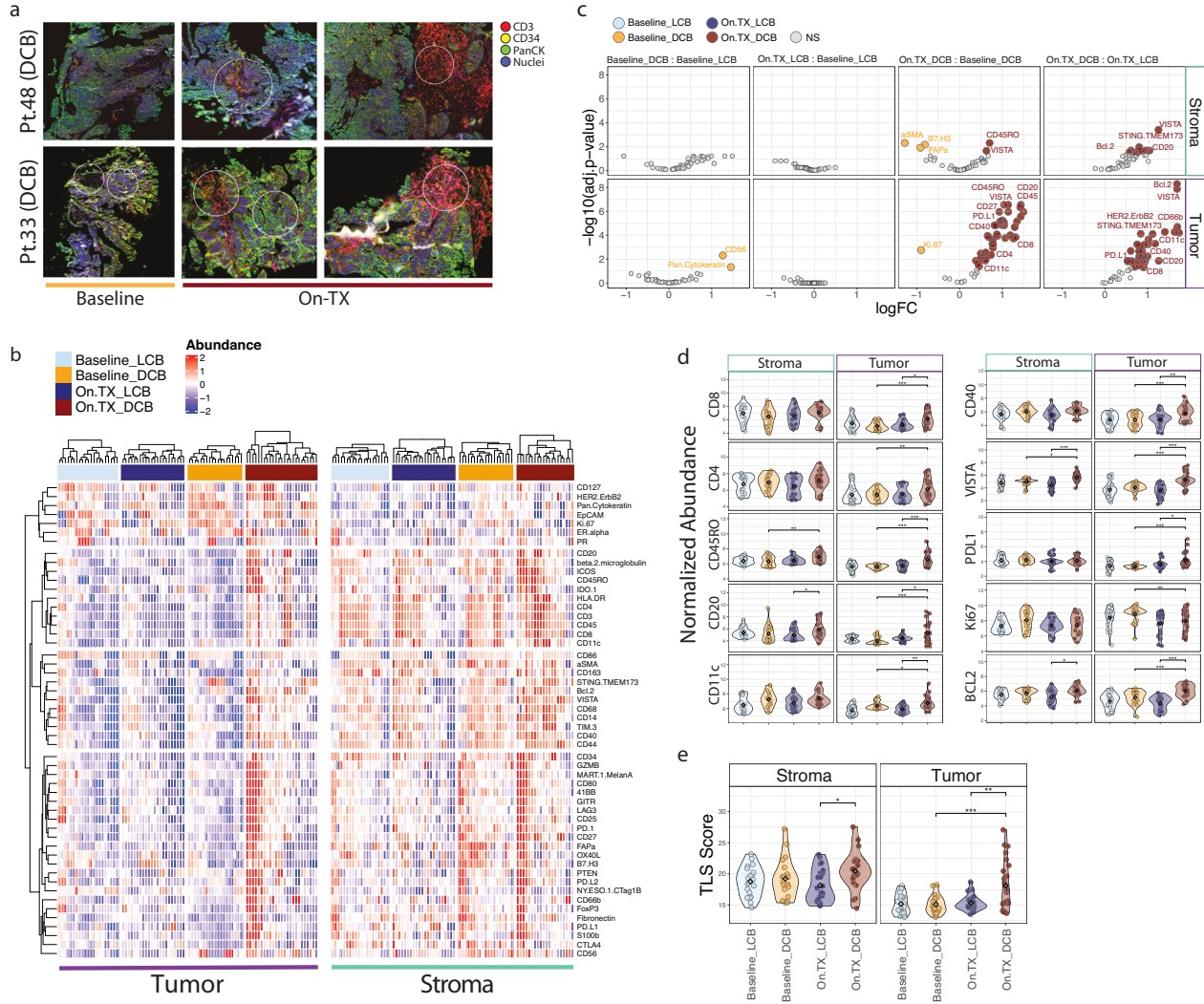

**Fig. 3 | Analysis of digital spatial profiling within tumor and stromal regions indicates patients with durable clinical benefit exhibit enhanced immune trafficking and activation compared to patients with limited clinical benefit.**
**a** Representative images from two DCB patients showing large CD3+ aggregates forming while On.TX. **b** Heatmaps representing all markers detected in the DSP assay in both tumor and stromal compartments. Tumoral immune marker abundances are increased with treatment in DCB (On.TX_DCB, red), as compared to both baseline DCB (Baseline_DCB, orange) and LCB baseline and on treatment (Baseline_LCB, light blue and On.TX_LCB, dark blue; respectively) patient populations. **c** Volcano plots demonstrate differential expression of these markers between our

Baseline and On.TX DCB and LCB patient populations, highlighting interesting biology like the appearance of Ki67 enrichment in Baseline_DCB, as compared to On.TX_DCB. **d** Quantifying immune and proliferative markers in the DSP data demonstrated significant differences in stromal and tumor fractions between important markers like PD-L1 and CD20 between On.TX_DCB and On.TX_LCB, suggesting altered immune phenotypes. *$p < 0.05$, **$p < 0.01$, Mann–Whitney U-Test. **e** TLS scores are significantly elevated in DCB patients On.TX relative to baseline DCB and On.TX LCB. *$p < 0.05$, **$p < 0.01$, ***$p < 0.001$; derived from linear regression of log-normalized abundances, by clinical group, correcting for the patient in paired analyses when appropriate, and multiple test corrections.

PFS (Supplementary Fig. 6b). The tumor microbiome demonstrated the same lack of alterations to alpha diversity, less microbial diversity, and limited overlap with the fecal samples, resulting in fewer differentially abundant microbes (Supplementary Fig. 7). To better understand how these bacterial populations may impact response to combination ICB, we next sought to profile the metabolic consequences associated with these varying bacterial populations.

**Metabolic alterations associated with response to combination ICB are enriched for amino acids and lipids**
We assessed global metabolism in serially collected fecal biospecimens to better understand the functional consequences of the differing bacterial compositions between DCB and LCB. Differential metabolite analysis revealed several specific metabolites in the fecal samples of DCB and LCB patients at baseline and On.TX (Fig. 4c, d).

Many key metabolites are associated with pathways that impact immune cell function, like bacterially-produced metabolites (indole and trimethylamine-N-oxide (TMAO))[30,31], fatty acids (FA:18:1)[32–35], acylcarnitine (C3-OH)[36], purine biosynthesis (Xanthine, Hypoxanthine)[37], and amino acids (serine and glutamate)[38]. In addition, fecal glutamate levels were markedly higher in DCB versus LCB On.TX (Fig. 4d).

In contrast, treatment similarly reduced indole levels in both DCB and LCB (Fig. 4d), indicating a potential function of glutamate in therapeutic response, which may represent a leverageable biological phenomenon[39,40]. Some serum metabolite levels (Supplementary Fig. 8) showed reciprocal patterns to those observed in fecal samples, like decreased phosphatidylcholines in the serum of DCB, compared to LCB at baseline, which may indicate increased excretion of these metabolites. Further, serum tryptophan metabolites (3-IAA), formed

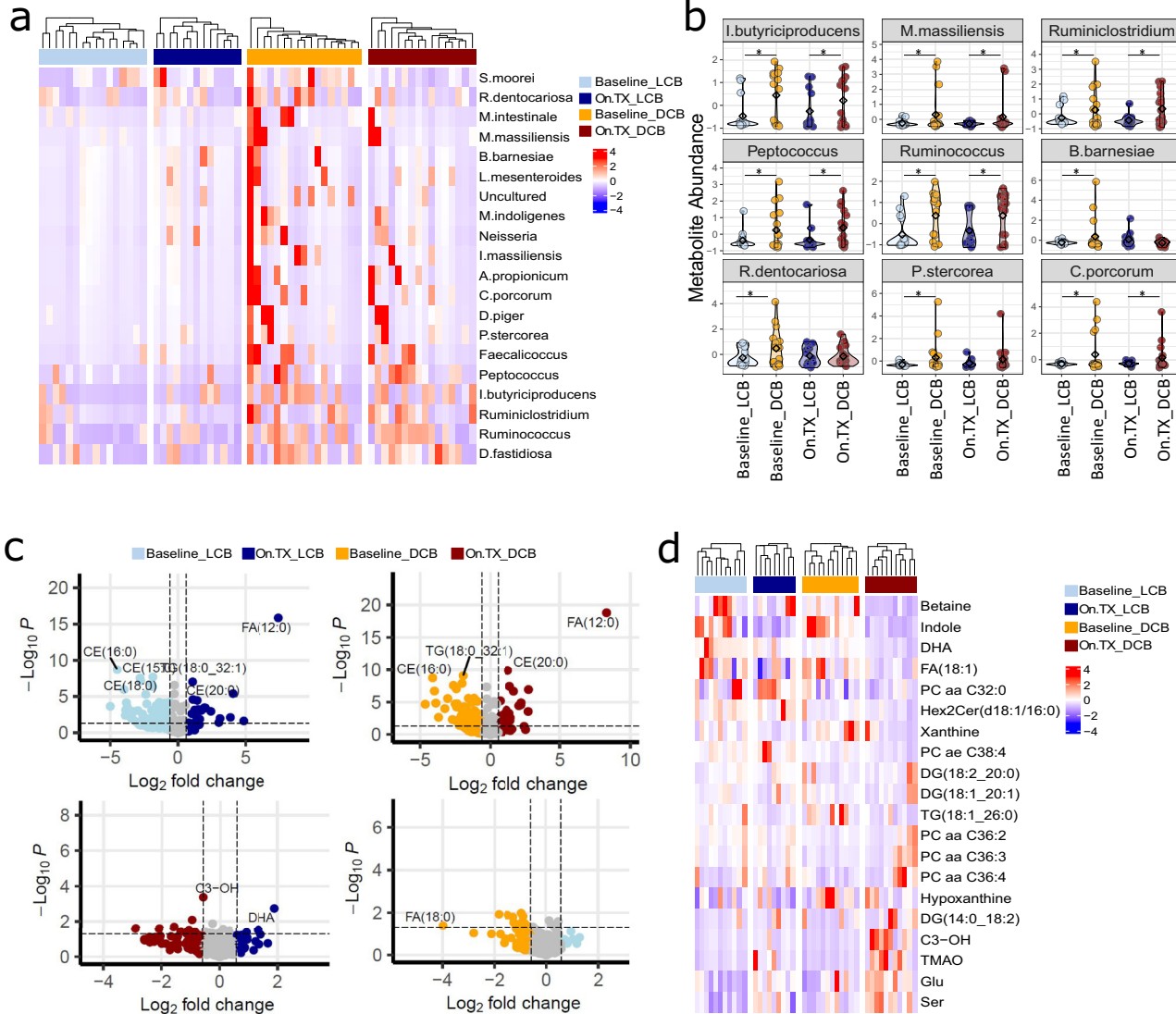

**Fig. 4 | Gut microbiome diversity and metabolic profile associated with durable responses to immunotherapy. a** Top 20 differentially abundant microbes (adj. p. value < 0.05, |LogFC|>1.5) between Baseline_DCB and Baseline_LCB displayed in a heatmap. Similar to tumor transcriptome, heatmaps demonstrate that across all microbes, patients with durable clinical benefit express higher microbe levels, and this expression is increased with treatment in DCB (On.TX_DCB, red), as compared to baseline DCB (Baseline_DCB, orange), baseline LCB (Baseline_LCB, light blue), and on treatment LCB (On.TX_LCB, dark blue) patient populations. **b** Violin plots quantify OTU abundance in all four groups for select bacterial species, all of which demonstrate higher levels of the bacterial species at baseline in DCB (Baseline_DCB, orange) as compared to LCB (Baseline_LCB, light blue), and more so with treatment in the DCB group (On.TX_DCB, dark red). *p < 0.05, **p < 0.01; derived from linear regression of log-normalized abundances, by clinical group, with multiple test

corrections. **c** Volcano plots demonstrate variety of significant (|logFC|>1.5, adjusted p < 0.1) differentially abundant metabolites between Baseline LCB (Baseline_LCB, light blue), Baseline DCB (Baseline_DCB, orange), On-Treatment LCB (On.TX_LCB, dark blue), and On-Treatment DCB (On.TX_DCB, red) patient populations. **d** Expression levels of the 20 most highly differentially expressed metabolites separate Baseline_DCB and Baseline_LCB fecal samples to a high degree based on Euclidean distances, across all 4 groups. Some metabolites were decreased with treatment in DCB and LCB (indole), some were solely increased in DCB with treatment (C3-OH, TMAO, Glu, and Ser), and some lipids were increased more so in LCB, regardless of timepoint (TGs and DGs). *p < 0.05, derived from linear regression of log-normalized abundances, by clinical group. Exact p-values can be found in Data Tables 2 and 3.

via indoles, were increased in DCB compared to LCB (Supplementary Fig. 8). Overall, many of the metabolites differentiating DCB from LCB belong to pathways relevant to immune function, especially those that are bacterially produced (TMAO[41] and indole[39]), which prompted us to explore in more detail associations between the metabolome, microbiome, and TME.

## Concurrent module analysis identifies potential key factors within the tumor-immune-gut axis linked to therapeutic response

Tumor metabolism is vital for tumor growth and progression and for supportive cell population function. This includes stromal and

vascular components and immune cell infiltration and function. To assess intrinsic tumor metabolism in the absence of tumor metabolomics, we used a bioinformatics pipeline previously developed to determine which of 114 metabolic pathways are most highly dysregulated based on transcriptomics-level data[42]. Metabolic prediction analysis from tumor transcriptome data (Fig. 5a) revealed several metabolic pathways that were highly dysregulated between DCB and LCB at both Baseline (top) and On.TX (bottom), including tryptophan metabolism and kynurenine metabolism, and several lipid-related pathways, consistent with dysregulation of indole and phosphatidylcholines, were observed from fecal metabolomics. To identify converged metabolic pathways linking the gut and TME, we combined

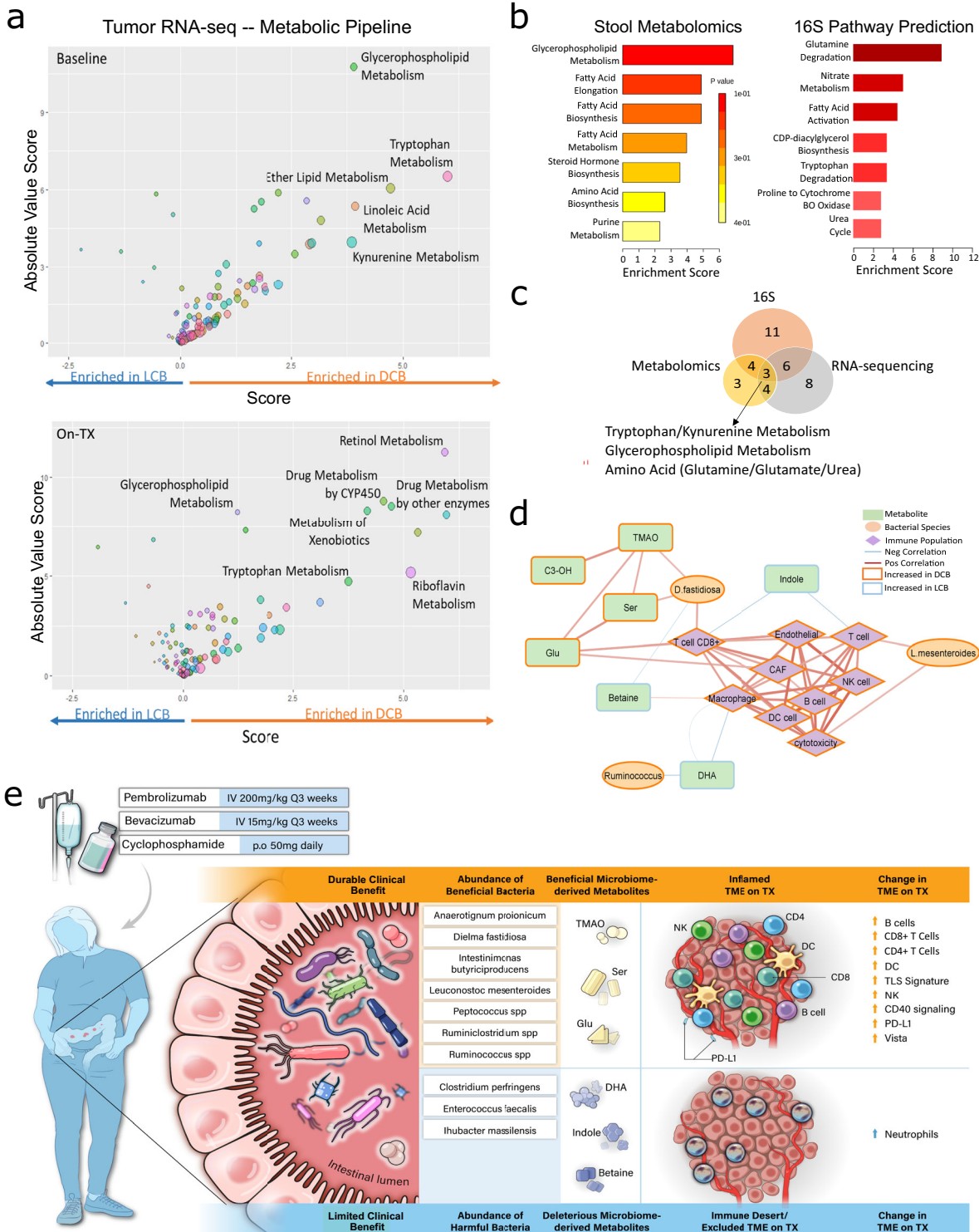

**Fig. 5 | Integration of bacterial species, metabolites, and immune populations identify points of convergence. a** Tumor transcriptomic data was utilized for metabolic pipeline assessment to understand which metabolic pathways were transcriptionally dysregulated significantly between the baseline DCB (orange) and LCB (light blue). Pathways like glycerophospholipid and tryptophan metabolism were among the most highly dysregulated pathways at both baseline (top) and on-treatment (bottom). **b** Significant differentially abundant metabolites from the same baseline comparison were used for Metabolite Set Enrichment Analysis (MSEA), highlighting altered glycerophospholipid metabolism (top left). Further, the differentially abundant OTUs from the baseline DCB vs. LCB analysis were functionally enriched to reveal alterations to similar pathways, like tryptophan/ amino acid metabolism, glycerophospholipids, and fatty acids (top right).

**c** Significantly dysregulated pathways identified between the tumor RNA-sequencing, stool metabolomics, and stool microbiome study overlapped, revealing 3 altered pathways common to all analyses. **d** Correlation network limited to metabolites (green rectangle), microbes (orange ellipses), and immune scores (purple diamond) nodes highly correlated with CD8 + T Cells, B cells, or Macrophages/ Monocytes ($R^2 > 0.39$ red), $R^2 < -0.39$ (blue), width of connectors increase with increased correlation value). Patient phenotype is designated by node borders with orange indicating the immune signature/microbe/metabolite was more highly expressed in DCB or blue for LCB. **e** Model demonstrating microbial (microbiome and metabolomic) and immune changes that underlie response to ICB in both DCB with DCB (top, orange) and LCB (bottom, blue) with no clinical response.

metabolic predictions from tumor transcriptomes with Metabolite Set Enrichment Analysis (MSEA) of fecal metabolomics (Fig. 5b, left) and functional taxonomic analysis (Tax4Fun) of the 16S sequencing (Fig. 5b, right). Overlap of enriched metabolic pathways between the fecal 16S, fecal metabolomics, and tumor transcriptomes (Fig. 5c) revealed three overlapping pathways, consisting of lipid (glycerophospholipid) and amino acid-related pathways (urea/glutamate and tryptophan metabolism). Similar to the fecal 16S, functional enrichment of the differentially abundant microbes from the comparison of baseline DCB and LCB resulted in the enrichment of the urea cycle and amino acids and inflammatory function (Supplementary Fig. 7d).

Fecal metabolite and microbiome abundances segregated into clusters positively and negatively correlated with immune deconvolution scores derived from tumor transcriptome data, indicating gut features that may be functionally associated with characteristics of the TME and response to combination ICB (Supplementary Fig. 9). One such metabolite was glutamate, which is involved in amino acid metabolism and sits at the crux of amino acid and carbohydrate metabolism via the urea cycle[43], correlating strongly with CD8 + T cell estimates in the TME. To identify global co-occurrence modules reflecting points of biological convergence that may explain differences in response to combination ICB, we built a correlation network linking TME signature scores, fecal metabolite levels, and fecal bacterial abundances (Fig. 5d, Supplementary Fig. 10). This network revealed a high number of significant correlations between metabolites, bacterial species, and tumor-infiltrating immune populations. For instance, we identified a highly correlated module linking serine and glutamate with *Dielma fastidiosa* in the gut to CD8 + T cells in the TME, which appeared to be driven by a combination ICB, as both *Dielma fastidiosa* and CD8 + T-cell estimates were highest in DCB On.TX (Supplementary Fig. 11, top), in contrast to other immune cell populations, like monocytes (Supplementary Fig. 11, bottom), where this relationship was not observed. Therefore, this module represents a significant node of convergence and may highlight a point of therapeutic leverage to enhance the efficacy of combination ICB, specifically targeting CD8 + T-cell fitness.

We pose a comprehensive working model (Fig. 5e) assessing durable responses of late-stage OC patients to a tolerable and highly efficacious combination ICB, which describes specific microbiota, like D. fastidiosa and Ruminococcus, and their metabolic outputs, like indole and DHA, in the gut supporting a unique TME. This study proposes that the immune milieu and host-microbiome may be tightly associated and further leveraged to stratify patients or enhance antitumor responses in future immunotherapy trials.

## Discussion

This study demonstrated that the combination of pembrolizumab, bevacizumab, and oral cyclophosphamide resulted in remarkable clinical responses in a subset of patients with recurrent OC. Notably, these patients experienced an approximately six month longer PFS than the typical median PFS expected for this cohort, and a superior QoL accompanied[14] this improvement compared to those undergoing multiple rounds of second-line chemotherapies. These clinical results now provide additional treatment options for women with recurrent OC[20].

While PFS and ORR were markedly improved, the biological basis for the response to this combination was not yet understood. We hypothesized that the gut microbiome and its related metabolic environment, which can be modified, might impact the OC tumor's immune environment, affecting the response to ICB[44–46]. This idea is primarily informed by prior research identifying links between the gut microbiome and varying responses to chemotherapy[47,48], immune modulation[11,49], and metabolic alterations[50,51]. Nonetheless, existing clinical and pre-clinical studies that have examined the microbiome's impact on immunotherapy response have yielded inconsistent

findings about the predictive value of specific bacterial species[52,53]. In addition, these studies often overlook the concept of functional redundancy in the microbiome, where different species can have similar effects, such as producing the same metabolite, within a specific ecosystem[54,55]. Our clinical trial provided a unique opportunity to assess the association of fecal microbiome and metabolic studies with immune modulation and clinical response.

Using a multi-omics analytical method, we examined clinical samples taken before and during treatment to understand the dynamic changes in the tumor, microbiome, TME, and immune system during our combination ICB therapy. We identified significant differences in the tumor transcriptome, immune populations, fecal microbiome, and fecal metabolome of patients who responded well to the treatment compared to those who did not. These findings suggest potential interventions, such as probiotics, selective antibiotics use, or fecal transplants, to enhance the effectiveness of immunotherapy in non-responsive OC patients[56].

This study's significant finding highlights the coordinated increase and movement of T and B cell infiltrates, suggesting TLS formation in DCB patients. In addition, the importance of both the humoral and cellular responses in improving the efficacy of combination ICB is supported by the enrichment of elevated T and B cell activation, differentiation, and proliferation signatures, as well as additional immune signatures associated with CD40, antigen presentation, cytokine production, and signaling, and indications of TLS. While aggregation of these cells in TLS may be challenging to identify in core biopsy samples, TLS transcriptional signatures[21,22] including CCL19, CXCL13, and CCL21 correlated strongly with clinical response to combination ICB, and tightly correlated abundances of T (CD3), B (CD20), and DC (CD11C) lineage markers were observed across 168 tissue regions and enriched in DCB patients, all pointing to TLS formation as a critical feature of durable response in our study.

The presence of functional TLS is known to improve responses to immunotherapies, making it an important focus for further research in late-stage OC patients[21,57]. Additionally, the co-regulatory molecules VISTA and CD40, which play a role in T-cell function and TLS formation, are identified as key features of response[58–60]. Responsive tumors also showed a reduction in the proliferative marker Ki67, indicating effective inhibition of tumor cell growth. To better comprehend the biology driving this enhanced immune state, we analyzed the gut microbiome and metabolome to identify links between bacterial and metabolic changes and immune cell accumulation in the TME.

Our research indicates that the fecal microbiome contributes to our patient's unique biological environment associated with treatment response. Through combined 16S RNA-sequencing and metabolomics profiling of patient fecal samples, we found that certain bacterial species, such as Ruminococcus and L.mesenteroides (Figs. 4 and 5), were notably present in responsive patients before and during treatment. In addition, these bacteria are known to produce lactic acid, a finding that aligns with previous studies, which suggest lactic acid from bacterial sources can enhance the response of both CD4+ and CD8 + T cells and improve the effectiveness of chemotherapy and immunotherapy by boosting cytokine production[61,62].

Certain dietary studies have explored this biological phenomenon, suggesting that increased consumption of certain foods, particularly nuts and walnuts[63], can boost these bacterial populations and enhance immunotherapy response[64]. However, less is understood about other bacteria linked with responsive patients, such as D.fastidiosa, especially in relation to immunotherapy and immune dysfunction. What little we know about D.fastidiosa suggests an association with amino acids, coenzyme A, and lipid metabolism[65]. This aligns with many of our metabolic findings (Fig. 4), where we observed increases in lipids like phosphatidylcholines and tri-/di-acylglycerides, amino acids like serine, and bacterial metabolites like TMAO in responsive patients.

Additionally, we observed decreases in metabolites like indole, a nutritional/bacterial compound, which supports the observed response differences. Indoles help balance the gut's anti-inflammatory properties by encouraging Treg differentiation and inhibiting Th17 cells[66]. To connect this to the tumor specifically, we performed a metabolic prediction analysis of the TME using transcriptome data. By comparing these predictions to MSEA and microbiome Tax4Fun profiling of the gut, we identified three key pathways: tryptophan metabolism, amino acid metabolism, and glycerophospholipid metabolism (Supplementary Fig. 9). This further underscores the interplay between the TME and the bacterial and metabolic composition of the gut. Changes, either increases or decreases in expression of key transcript and metabolites within the tryptophan metabolic pathway, in which indoles are key players[67], can influence immune cell function and alter the response to immunotherapies[68]. Among the many positive links between bacterial species, immune cell populations, and metabolic abundance, one noteworthy association includes D.fastidiosa and related metabolites like bacterial TMAO and the amino acids serine and glutamine, all of which show a strong correlation with CD8 + T cells (Fig. 5b). The microbial metabolite TMAO plays a crucial role in responding to ICB. In fact, administering TMAO directly into the abdominal cavity or via a dietary choline supplement in mice with pancreatic tumors reduced tumor growth by enhancing the T cell response in the TME[41]. This suggests that dietary interventions could help non-responsive patients become responsive, depending on their microbial composition (D.fastidiosa) and metabolic environment (supplementation of TMAO and choline or glutamine).

Although this study involves a relatively small group of participants ($n = 40$), it provides considerable insights into the gut microbiome's influence on OC treatment. Further, there are considerable challenges when assessing the microbiome via 16S rRNA-sequencing, and extrapolating function from taxonomy. Additionally, this method lacks in terms of specific enzymatic function and expression, resulting in only an ability to assess correlations between specific bacterial OTUs and function, which is where metatranscriptomics techniques succeed. However, our utilization of 16S functional assessment, overlapped with tumor transcriptomics and metabolomics data, strengthens confidence in the findings of this data. This underscores the potential of this research area and emphasizes the need for more expansive studies with larger patient groups for more comprehensive validation. Future research should explore the nuances of individual drugs, bacterial species, the shared functions across different species, and the complex interplay among bacterial metabolic activities, the immune system, and immunotherapy response. This approach will build on this study's foundational work in uncovering these intricate relationships.

The results of our investigation highlight the pivotal importance of B cells, TLS, and the gut microbiome, along with its metabolic milieu, in augmenting the efficacy of combination ICB therapy in OC. These insights pave the way for innovative therapeutic interventions. For instance, exploring additional combination therapies, such as those involving CD40 agonist antibodies (e.g., the ongoing NCT05231122 trial), modifying dietary patterns, or incorporating probiotic supplements. Such strategies have the potential to significantly improve the outcomes in patients facing advanced-stage or recurrent OC, offering new avenues to enhance survival and QoL.

## Methods
### Clinical trial overview
This investigator-initiated (I-270715), single-arm, phase 2 non-randomized clinical trial (NCT02853318) of pembrolizumab and bevacizumab with oral cyclophosphamide enrolled 40 patients from September 6, 2016, to June 27, 2018, with clinical data collected until November 1, 2021. This study was approved by the Roswell Park Institutional Review Board (IRB), and all patients provided written informed consent before initiating any study procedures. Participants

did not receive financial compensation. This study followed the Consolidated Standards of Reporting Trials (CONSORT) reporting guideline. Patient eligibility, patient demographics, study design, and treatment were previously described[14].

### Statistics and reproducibility
Clinical data were analyzed from September 6, 2016, to November 1, 2023. With an accrual goal of 40 evaluable patients, the study had 86.1% power to detect an improvement in the 7-month PFS rate from 0.3 to 0.5 (at $\alpha = 0.10$). Therefore, the null hypothesis is based on the presumed activity of standard chemotherapy, single-agent bevacizumab, or a combination of low-dose oral metronomic cyclophosphamide and bevacizumab in recurrent OC, with a median duration of response of 3.9–4.5 months[7,13]. An improvement to a median PFS of 7.0 months (i.e., a 7-month PFS rate of 0.5) would be considered significant in this study population.

The analysis includes all patients who received study treatment. The PFS and OS were summarized using Kaplan–Meier methods, where in estimates of the median were obtained with 90% CIs. The response was summarized using frequencies and relative frequencies. No adjustments were made for multiple testing. All clinical analyses were performed in SAS, (version 9.4; SAS Institute, Inc), at a two-sided significance level of $\alpha = 0.10$ and visualized in R (3.6.1).

### Biospecimens
Tumor biopsies were obtained at baseline and after 3 cycles of treatment (On-TX) on cycle 4 day 1. Blood and fecal samples were collected at baseline, after 3 cycles of treatment, and after 9 cycles of treatment. Seventy one fecal samples were collected according to the International Human Microbiome Standards (IHMS) guidelines, and were submitted for 16S rRNA gene sequencing for microbiome analysis, and 42 fecal samples were submitted for comprehensive metabolomics profiling. Digested tumor samples from 29 patients were submitted for bulk-RNA-sequencing, respectively and 28 samples from 14 patients were analyzed via Digital Spatial Profiling (DSP). The investigators were not blinded to allocation during experiments and outcome assessment, to ensure LCB and DCB patients were being compared. Therefore, subsequent experiments were not randomized.

### RNA-sequencing
Samples were delivered on dry ice to the Roswell Park Genomics Shared Resource (GSR) for RNA extraction and RNA-seq analysis after complete collection for all participants. Purified RNA was prepared using the miRNeasy micro kit (Qiagen, Cat No. 217084), and sequencing libraries were prepared with SureSelect XT RNA Direct kit (Agilent Inc) from 200 ng total RNA and hybridized to the SureSelectXT Human All Exon V6 + UTR Capture library (Agilent Inc.). The final RNA-seq libraries were sequenced on Illumina NovaSeq 6000 using 100 paired-end sequencings. Bioinformatics pre-processing and quality control (QC) steps were carried out by the Roswell Park Bioinformatics Shared Resource using an established pipeline following commonly adopted practices for RNA-seq data analysis. Raw reads that passed the Illumina RTA quality filter were demultiplexed and pre-processed using FastQC for sequencing base quality control. Reads were then mapped to the latest version of a human reference genome (GRCh38) and reference transcriptome GENCODE (v25)[69] using Bowtie (v1.0.1)[70] and TopHat (v2.0.13)[71] aligner. Mapped reads were quantified at the gene level as a raw counts matrix using featureCounts from Subread[72] using fracOverlap 1 (only entire reads overlapping to annotation feature are counted). Samples were filtered after quality assessment, and a total of 48 samples were used in subsequent analyses (Baseline_DCB; $n = 13$, Baseline_LCB; $n = 14$, On-TX_DCB; $n = 8$, On-TX_LCB; $n = 13$). Raw feature counts were normalized and differential expression analysis was carried out using DESeq2[73]. Differential expression rank order was

used for subsequent gene set enrichment analysis (GSEA)[74], performed using the cluster profile package in R. Gene sets queried included the Hallmark, Canonical pathways, and GO Biological Processes Ontology collections available through the Molecular Signatures Database (MSigDB)[75]. Gene signatures reflecting the presence of tertiary lymphoid structures (TLS) were compiled from published papers (Chai-semartin et al.[22], Cabrita et al.[21]) or curated from various sources (*CXCL13, CXCR5, CCL19, CCL21, LAMP3, BANK1, CR2, MS4A1, PAX5, BCL6, FCER2, NTAN1, CD4*). Single-sample GSEA (ssGSEA) scores were calculated using the GSVA package. Immune cell type deconvolution scoring using 6 estimation methods was performed using TIMER[76]. Liver contamination was quantified via ssGSEA utilizing a hepatocyte specific gene list derived from[77], and included as a covariate in the final model design for differential expression determination.

## Digital spatial profiling

Fifty-two distinct proteins were chosen that assess samples' immune milieu, proliferative capacity, and tumoral phenotype. The slide preparation for the GeoMx Human IO (Immuno-Oncology) assay (Nano-String) followed standard procedure. Briefly, fresh frozen paraffin embedded (FFPE) human OC samples were deparaffinized sequentially with CitriSolv and ethanol. Samples then underwent antigen retrieval in Citrate Buffer (pH6.0) for 15 min. at 100 °C. After blocking, slides were incubated overnight with the GeoMx IO assay antibody cocktail and morphology markers. The following day, slides were washed and loaded onto the GeoMx prototype machine for DSP. On the GeoMx device, after hybridization of specially oligo-conjugated antibodies to a section, the tissue is counterstained and imaged using immuno-fluorescence microscopy. Structures of interest are visualized and selected as Regions of Interest (ROI). An ultraviolet (UV) laser pulse focuses on this ROI to release oligos conjugated to the target proteins or RNAs. The DSP instrument aspirates these oligos containing barcodes into collection wells using a capillary micropipette. The barcodes map them to their source protein, and digital counting of these oligos with nCounter (Nanostring) permits a direct readout of target abundance localized to the specific ROI within the tissue section. The NanoString GeoMx DSP allows direct quantification of proteins to spatially resolved anatomic structures in tissue sections. Counts were normalized to a panel of housekeeping genes and log transformed before statistical analysis. Principal component analysis was performed to identify latent features most associated with response groups (PC5) and region type (PC1). Differential abundance was performed on normalized abundance counts using Limma[78]. TLS score was based on summed signal of T (CD3), B (CD20), and DC (CD11C) markers.

## 16S-RNA sequencing

DNA from tumor (Total tumor $n = 34$; tumor baseline $n = 17$, tumor On-TX $n = 17$) and fecal (Total fecal $n = 78$; fecal baseline $n = 45$, fecal On-TX $n = 33$) samples was extracted in batches, with randomization of samples, but sequenced together at the GSR. DNA extraction was performed using Qiagen DNeasy PowerSoil Pro Kit following manufacturer's recommendations, and the sequencing libraries were prepared using a two-step PCR method for targeting a -500 bp region of the 16S V3 and V4 rDNA. In the first PCR, 25 ng of DNA is used to amplify the target region, where the PCR primers have overhang adapter sequence necessary for the second PCR step. After purification, the amplicons from the first PCR were indexed using the Nextera Index Kit (Illumina Inc.) for pooling of libraries and multiplex sequencing. Purified libraries were then validated using Agilent Tapestation 4200 D1000 tape (Agilent Technologies); validated libraries were pooled equal molar in a final concentration of 4 nM in Tris-HCl 10 mM, pH 8.5, before 2 ×300 cycle sequencing on a MiSeq (Illumina, Inc.) using the appropriate v3 reagents[79]. The median read count was 350,792, ranging from 2999 to 550,337. Paired-end fastq

reads were demultiplexed, processed and analyzed using QIIME v1.9.1[80]. Operational taxonomic units (OTUs) are then assigned using QIIME's uclust-based open-reference OTU-picking pipeline using SILVA 16S rRNA reference (v132)[81]. Bacterial sequences were aligned using PyNAST[82], and refined using ChimeraSlayer[83]. OTUs with less than 0.001% assigned sequences were removed from each sample to avoid biased and inflated diversity estimates. Positive and negative control samples were checked against the whole batch and then removed from the data. OTUs were collapsed at the genus level. A pairwise alignment-based quantification method, BLCA[84], was employed to classify reads at the species level. Using BLASTN[85] (v2.9.0), amplicon reads were queried against NCBI's microbial sequence and taxonomic databases. From BLCA's output, reads with a minimal confidence score of 85 at the species level were kept. Observed, Chao1, Shannon, and Simpson's Reciprocal diversity indices, were estimated for alpha-diversity scores using the phyloseq package (v1.28.0)[86]; these are mean estimates performing 100 boot-strapped rarefactions. Diversity group comparisons were performed using two-sample t-tests. For beta-diversity, Bray–Curtis dissimilarity score paired with classical multidimensional scaling was estimated and tested using the PERMANOVA[87] procedure (3000 permutations) implemented by the vegan package (v2.5.6)[88]. Additional summary statistics, tests, and visualization for the alpha and beta diversity were performed comparing outcomes of interest, including Class and Phylum - composition plots. Statistical analyses and comparisons were conducted to detect microbial differential abundance[89] using Limma[78] (v3.54.2) at the genus and species level. log2Fold-Change and heatmap plots were used to examine relevant OTUs subsetting those having an FDR adjusted $p$-value < 0.05 (Supplementary Data 2). Functional taxonomic analysis was carried out using the MicrobiomeAnalyst[90] software for marker data profiling (MDP), and the embedded Tax4Fun[91] functional analysis. KEGG[92] enzymatic ids were then analyzed using DESeq2, to identify differential bacterial genes, and enriched for metabolic processes. Statistical analyses were performed in R (3.6.1) and related dependencies.

## Metabolomics

Fecal and serum samples were prepared and analyzed in the Roswell Park Comprehensive Cancer Center Bioanalytics, Metabolomics and Pharmacokinetics (BMPK) Shared Resource, using the MxP Quant 500 kit (Biocrates Life Sciences AG, Innsbruck, Austria) in accordance with the user manual. Human fecal samples were homogenized in an 8-fold volume of 75% ethanol using optimized settings on the Omni-Bead Ruptor 24 (Omni International, Kennesaw, GA). The homogenate (80 μL) was vortexed, sonicated, mixed with 600 μL of MTBE, and then shaken for 1 h. Samples were mixed with 150 μL of water and centrifuged. The entire supernatant was transferred to a clean tube, evaporated to dryness, capped, and stored at −80 °C until the day of the kit analysis. To reconstitute on the day of kit analysis, 40 μL of isopropanol (IPA) was added to each sample followed by 40 μL of 30% IPA with vortexing after each addition. Samples were centrifuged to obtain a supernatant. 40 μL of each supernatant (in two 20 μL additions, dried under nitrogen between additions) or 10 μL of each quality control (QC) samples, blank, zero sample, or calibration standard were added on the filterspot (already containing internal standard) in the appropriate wells of the 96-well plate. The plate was then dried under a gentle stream of nitrogen. The samples were derivatized with phenyl isothiocyanate (PITC) for the amino acids and biogenic amines and dried again. Sample extract elution was performed with 5 mM ammonium acetate in methanol. Sample extracts were diluted with either water for the HPLC-MS/MS analysis (1:1) or kit running solvent (Biocrates Life Sciences AG) for flow injection analysis (FIA)-MS/MS (50:1), using a Shimadzu HPLC system interfaced with a Sciex 5500 mass spectrometer (MS). Data was processed using MetIDQ software (Biocrates Life Sciences AG), and Limma[78] for differential metabolite

analysis. Differential metabolites were then enriched (MSEA) using the MetaboAnalyst software. Correlation networks were generated using the "corrplot" and "igraph" packages in R[93]. Statistical analyses were performed in R (3.6.1) and related dependencies.

## Co-occurrence module integrative analysis

Venn diagrams were utilized to determine which metabolic pathways were commonly associated with fecal microbiome, fecal metabolome, and tumor transcriptomic data, at both baseline and on-treatment. Co-occurrence modules were then assessed using correlation networks. A correlation matrix was generated using the corr package in R for Immune Deconvolution scores (tumor bulk RNA-sequencing), metabolomics (fecal samples), and bacterial species (fecal 16S RNA-sequencing samples, Baseline DCB vs. LCB). These results were then visualized as a network diagram, where each entity of the dataset was represented by a node, and 2 nodes were connected if their correlation or distance reached a threshold (0.4). To make a graph object from the correlation matrix, we used Cytoscape. Node features were mapped to the data source variable (e.g. tumor transcript, fecal microbiome, fecal metabolome). This gives an additional layer of information, allowing for the comparison of the network structure with a potential expected organization. When appropriate, an FDR corrected $p$ value < 0.05 was utilized as a cut off.

## Reporting summary

Further information on research design is available in the Nature Portfolio Reporting Summary linked to this article.

## Data availability

Source data are provided within this paper, and in publicly available databases. Processed data derived from RNA-seq (GSE206422) are deposited at the NCBI Gene Expression Omnibus. Data from 16S rRNA-seq (PRJNA852556) are deposited at the NCBI Sequence Read Archive. Metabolomics data can be found in (Supplementary Data File 3).

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

## Acknowledgements

This work was supported by National Cancer Institute (NCI) grant P30CA016056 involving Roswell Park Comprehensive Cancer Center's Genomics Shared Resource, Bioinformatics Shared Resource, and Bioanalytics, Metabolomics, and Pharmacokinetics Shared Resources. The research reported in this publication was supported by the NIH Office of Research Infrastructure Programs under award number S10OD024973; the Center for Computational Research at the University at Buffalo maintained the computational resources. Work in this grant was also supported by The Roswell Park Alliance Foundation (EZ), grant funding (52622) from Merck and Company (E.Z.), U24 CA232979-01S5 (S.R.R., M.D.L., S.L.), R25CA203650 (S.R.R.), and R50CA283805 (P.K.S.).

## Author contributions

S.R.R. contributed to manuscript writing and editing, microbiome and metabolome studies, bioinformatics data analysis, visualization, and interpretation; M.D.L. contributed to manuscript writing and editing, transcriptome and DSP studies, and bioinformatics data analysis, visualization, and interpretation. E.C.G. contributed to bioinformatics processing, data analysis, and graphic representation; S.C. contributed to data discussions; S.B. and J.W. contributed to data and bioinformatics discussions; P.K.S. contributed to genomics study design and implementation including RNA-seq, DSP, and 16S RNA bacterial studies; K.W. contributed to biostatistical analysis; K.A. contributed to study design, biostatistical analysis, and manuscript preparation; S.M.H. contributed to data analysis, manuscript writing and editing, and submission; A.J.R.M. contributed to intellectual discussion of data interpretation and manuscript review; K.O., B.H.S., and G.P. contributed to the design of the clinical trial, translational work, and manuscript review; S.L. contributed to bioinformatics data discussion; J.A.W. contributed to intellectual discussion of data interpretation and manuscript review; E.Z. contributed to designing and executing the clinical trial, securing funding for the project, data analysis, and manuscript writing, editing, and submission. All authors were involved in the critical review of the manuscript and approved the final version for submission.

## Competing interests
The authors declare no competing interests. This study was approved by the Roswell Park Institutional Review Board (IRB), and all patients provided written informed consent before initiating any study procedures. Participants did not receive financial compensation. This study followed the Consolidated Standards of Reporting Trials (CONSORT) reporting guideline. While all work was completed at the Roswell Park Comprehensive Cancer Center by author Shanmuga Chilakapata, this author has relocated to Northeastern University.
