## [Transparent Peer Review file · Nature Communications]

Integrative Multi-Omics Analysis Uncovers Tumor-Immune-Gut Axis Influencing Immunotherapy Outcomes in Ovarian Cancer

Corresponding Author: Dr Emese Zsiros

This manuscript has been previously reviewed at another journal. This document only contains reviewer comments, rebuttal and decision letters for versions considered at Nature Communications.

Version 0:

Reviewer comments:

Reviewer #1

(Remarks to the Author)

The authors have addressed almost all of my previous concerns.

Figure 2: I particularly appreciate the change in language when describing Figure 2. Certainly, the GSEA data in Figure 2a support the interpretation that T- and B-cell signatures are higher in Ov TX_R compared to baseline_R. In contrast, the other analyses in Figure 2 show no apparent differences between baseline_R and On TX_R, which does not accurately support the statement that there are elevated mean enrichment scores for T and B cell signatures on treatment in the responder group, but this is a minor point.

Methods, Line 18: Need to put the references in parentheses. Although it is indicated that this has been corrected, in actual fact it has not in the manuscript version that was provided.

Reviewer #3

(Remarks to the Author)

Thank you for addressing many of the points raised in the initial review. I have a few additional comments:

1. ORR Analysis:

o I appreciate the importance of duration of response/PFS, and desire to factor that as a response. One of the challenges in a small study is to reduce outcome variables and having a binary definition of R and NR makes correlations easier, but there needs to be a caveat that this is not a 'generally accepted' definition of response/responders and was developed by a 'consensus of the investigators'. There are international recommendations from groups like GCIG which have definitions based on formally defined and published criteria (Ovarian Cancer Consensus recommendations for Clinical Research). I am not taking issue of benefit authors report in long term responders, but best way of incorporating this into the correlations.

o It would be informative to see the results based on the objective ORR and if there are different patterns seen in patients who have objective regression vs prolonged stabilization. Current guidelines, including the reference cited (PMID: 33211063), recommend randomization for studies using PFS as a primary endpoint, ORR is often considered the most appropriate endpoint for single-arm studies.

2. Clarification on durable stable disease:

o The explanation for including patients with PFS beyond the median of 10.2 months as responders is well-justified, particularly in the context of comparing expected outcomes with other treatments. Highlighting the clinical significance of durable stable disease is important and important to include in the m/s.

3. Correlation between multi-omics analysis and clinical outcomes:

o It would be interesting to assess correlate multi-omics data with conventional clinical parameters for response – though the

numbers may be too small. Among the 8 responders who were platinum-sensitive, are there any discernible metabolic, molecular, or microbiome differences?

o

4. Prospective assessment of measures:

o As there is no data on quality of life, in discussion (P14, lines 409-413), be state that these measures require prospective validation and assessment before concluding that they could potentially improve outcomes and quality of life.

Reviewer #5

(Remarks to the Author)

The authors of this study used an integrated multi-omics approach to unravel the effect of tumor microenvironment and host-microbiome to enhance antitumor responses for future immunotherapy trials in recurrent ovarian cancer patients. They showed that patients who responded well to the combination ICB therapy has enhanced tumoral immune cell infiltration at baseline and after treatment compared to non- responders. In addition, they reported that the fecal microbiome and metabolome, might impact the OC tumor's immune environment, affecting the response to ICB.

The manuscript is well written, figures are self-explanatory and supplementary data is highly supportive of the observations. However, a few of my concerns are listed below:

Comments:

Introduction- Line 63-64: Reference not formatted

In supplementary Fig1a, authors have observed that BRCA-positive patients had a 71.4% ORR compared to 30.4% in BRCA-negative patients ($p = 0.02$). What could be the possible reason behind this observation?

I understand that the authors have noticed abundance of some microbiome species in responders while some other in non-responders and the presence of several species was altered upon exposure to therapy. Firstly, is fecal sample the right biological specimen for studying differences in microbiome of ovarian cancer patients who respond to therapy versus non-responders. Secondly, how would you relate these findings in terms of biological cause and further implementation in treatment intervention?

As this finding has a translational value, what would be future work in this direction?

Version 1:

Reviewer comments:

Reviewer #3

(Remarks to the Author)

The Authors have addressed most of my concerns. I think the 'internal' consensus to call prolonged SD as responders is not validated or widely accepted, and remains a challenge. I understand the reasons for doing this and the lack of an adequate accepted measure which captures prolonged disease control - but in my opinion the powerful results stand by themselves with validated endpoints.

That having been said, I think the results are important, and provocative, and warrant further investigation in future studies to modulate the TME/Microbiome/Metabolomic interactions.

Reviewer #5

(Remarks to the Author)

Thank you for comprehensively addressing all of the comments and critiques of the reviewers.

Dr. Michael Dean

Nature Communications Response to Reviewers

REVIEWER COMMENTS are in black. AUTHOR COMMENTS in blue.

Reviewer #1 (Remarks to the Author):

- The authors have addressed almost all of my previous concerns.
- Figure 2: I particularly appreciate the change in language when describing Figure 2. Certainly, the GSEA data in Figure 2a support the interpretation that T- and B-cell signatures are higher in Ov TX_R compared to baseline_R. In contrast, the other analyses in Figure 2 show no apparent differences between baseline_R and On TX_R, which does not accurately support the statement that there are elevated mean enrichment scores for T and B cell signatures on treatment in the responder group, but this is a minor point.

Response: We appreciate this concern. In the text, we have emphasized there are differences with multiple cell types aside from B and T cells, in Figures 2c and 2d.

- Methods, Line 18: Need to put the references in parentheses. Although it is indicated that this has been corrected, in actual fact, it has not in the manuscript version that was provided.

Response: We have generated an entirely new set of citations, specifically for the methods you are referencing, now. They can be found within the methods document, and all reporting has been updated and standardized.

Reviewer #3 (Remarks to the Author):

- Thank you for addressing many of the points raised in the initial review. I have a few additional comments:

1. ORR Analysis:

- I appreciate the importance of duration of response/PFS, and desire to factor that as a response. One of the challenges in a small study is to reduce outcome variables and having a binary definition of R and NR makes correlations easier, but there needs to be a caveat that this is not a 'generally accepted' definition of response/responders and was developed by a 'consensus of the investigators'. There are international recommendations from groups like GCIG which have definitions based on formally defined and published criteria (Ovarian Cancer Consensus recommendations for Clinical Research). I am not taking issue of benefit authors report in long term responders, but best way of incorporating this into the correlations.

- It would be informative to see the results based on the objective ORR and if there are different patterns seen in patients who have objective regression vs prolonged stabilization. Current guidelines, including the reference cited (PMID: 33211063), recommend randomization for studies using PFS as a primary endpoint, ORR is often considered the most appropriate endpoint for single-arm studies.

Response: Thank you for this insightful comment. We have had extensive discussions with our co-authors both before conducting the analyses and while revising the manuscript in response to these concerns. Below are the key points addressing your feedback:

- While the GCIG guidelines are indeed valuable for designing clinical trials in ovarian cancer, they do not adequately address the unique challenges of immunotherapy translational research. Specifically, there is no guidance on defining responders (R) versus non-responders (NR) in the context of immunotherapy, a challenge faced by our study and many others globally. This gap suggests that an update to the GCIG guidelines to include considerations for immunotherapy trials and translational research may be warranted, and we plan to raise this issue at the next GCIG meeting. A recent paper published on this by Saad et al (PMID: 37760636) is an excellent summary of “Methodological Toolkits” in immuno-oncology (IO) clinical trial designs, which lacks in the field of Gynecologic Oncology.
- Although PFS and ORR are accepted endpoints for phase 2 trials, the goal of this paper is not to re-report the clinical trial results already published in JAMA Oncology (PMID: 33211063), which used traditional definitions to report clinical trial outcomes, but to provide updated and final clinical data on these patients and also to identify various factors in patients that drove the most clinical benefit from this trial for future studies.
- As noted in our previous response, immuno-oncology trials involving patients with long-term stable disease present a challenge. In cancers where median PFS is only 3-4 months for any second-line single agent, long-term stable disease should not be categorized as treatment failure. Labeling such patients as NR for translational work could be misleading and hinder our ability to distinguish between those with favorable versus unfavorable outcomes.
- To further illustrate this, we present survival curves in our study comparing patients grouped by durable clinical benefit (Top panels) versus those grouped solely by ORR (Bottom Panels). The separation of survival curves is much clearer when using the durable clinical benefit grouping. The top two panels demonstrate the distinction in PFS/OS when selecting patients with durable clinical benefit, while the bottom two panels show the less distinct separation when using ORR alone.

Additionally, in our JAMA Oncology paper, we demonstrated that patients with stable disease and no increase in disease burden, compared to those with partial response PR or CR, showed no significant difference in PFS/OS, suggesting that durable SD is a meaningful clinical and immunological response in OC immunotherapy trials.

Figure legend from JAMA Oncology paper:

Among patients with a 0% to 30% decrease in tumor burden, the median 6-month PFS rate was 0.73 (90% CI, 0.45-0.89); the median PFS was 10.3 (90% CI, 4.3-27.1) months. Among those with more than 30% to 100% decrease in tumor burden, the 6-month PFS rate was 0.94 (90% CI, 0.72-0.99); median PFS, 17.4 (90% CI, 7.8-24.4) months (log-rank $P = .47$). (PMID: 33211063)

- Upon reflecting on your valuable comments, we realize that the confusion likely arises from our use of the terms "Responders" and "Non-responders." To avoid any ambiguity, we recognize the need to revise our terminology. Our intent was to highlight the cohort of patients who derived the most clinical benefit from the treatment combination, rather than strictly adhering to ORR definitions, which are more commonly applied to chemotherapy or targeted therapy trials.
- To address this, we have revised the manuscript and relabeled our patient groups as **"Durable Clinical Benefit"** (DCB) versus **"Limited Clinical Benefit"** (LCB). This change, along with additional clarification in the text, ensures that our focus is clear: identifying key factors that can lead to long-term/exceptional responses and survival benefits in IO clinical trials in OC patients and guide the design of future clinical trials. This is also clearly stated throughout the manuscript. Specifically, we aim to replicate the deep and lasting responses observed in our OC patients, which the traditional ORR definition fails to fully capture. Our revised approach emphasizes practical insights for identifying and optimizing such high-benefit patient groups in subsequent studies. We trust this clarification not only addresses your concerns, but also reinforces the alignment of our approach with the objectives of this translational paper.
- Methods section correction: *In addition to patients achieving a confirmed partial or complete response as per iRECIST criteria in the clinical trial, individuals with PFS beyond the median of 10.2 months were classified as patients with durable clinical benefit (DCB) for the correlative analyses. This decision stems from the understanding that sustaining durable, stable disease in an immunotherapy trial signifies a considerable clinical benefit¹⁷, especially for this cohort of patients, where the anticipated median PFS for second-line chemotherapy stands at 3-4 months and 1.9-2.1 months for pembrolizumab as a single agent¹⁸⁻²⁰.*

2. Clarification on durable stable disease:

- The explanation for including patients with PFS beyond the median of 10.2 months as responders is well-justified, particularly in the context of comparing expected outcomes with other treatments. Highlighting the clinical significance of durable stable disease is important and important to include in the m/s.

Response: Please see the answer above and we edited the manuscript to highlight this.

3. Correlation between multi-omics analysis and clinical outcomes:

- It would be interesting to assess correlate multi-omics data with conventional clinical parameters for response – though the numbers may be too small. Among the 8 responders who were platinum-sensitive, are there any discernible metabolic, molecular, or microbiome differences?

Response: Thank you for your comment. This study did not specifically investigate immune, microbiome, or metabolic profiles in platinum-resistant versus platinum-sensitive patients due to the limited number of platinum-sensitive patients enrolled in the trial.

- However, we were also intrigued by this finding and did additional analysis focused on the number of prior lines of therapy and the evaluation of peripheral blood samples from all 40 participants. We particularly assessed senescent and exhausted T cells in circulation, hypothesizing that prolonged exposure to cytotoxic chemotherapy, especially alkylating agents like platinum, might induce premature "aging" of the immune system. This aging could potentially influence the efficacy of immunotherapy in platinum-resistant patients, who generally have had more extensive exposure to these treatments.
- Our findings indicated that DCB to the triple regimen had a significantly lower number of circulating senescent T cells compared to LCB. However, the number of prior therapy lines alone did not predict response effectively. Factors such as age and BRCA mutation status also correlated with the degree of immune senescence, which aligns with expectations.
- These observations were presented at the Society of Gynecologic Oncology Annual Meeting in 2023 (Gaulin N et al.) and are being further examined in a separate manuscript that explores biomarkers for predicting responses to immunotherapy combinations. This ongoing work underscores our commitment to deepening the understanding of how underlying patient characteristics and prior treatments influence therapeutic outcomes.

4. Prospective assessment of measures:

- As there is no data on quality of life, in discussion (P14, lines 409-413), be state that these measures require prospective validation and assessment before concluding that they could potentially improve outcomes and quality of life.

Response: The study used EORTC questionnaires to assess the QoL of the patients throughout the clinical trial. These quality-of-life survey results were all published in our JAMA Oncology paper (PMID: 33211063). This has been further clarified in the text (Introduction, line 87) and the reference added. (Ref. 14)

Reviewer #5 (Remarks to the Author):

- The authors of this study used an integrated multi-omics approach to unravel the effect of tumor microenvironment and host- microbiome to enhance antitumor responses for future immunotherapy trials in recurrent ovarian cancer patients. They showed that patients who responded well to the combination ICB therapy has enhanced tumoral immune cell infiltration at baseline and after treatment compared to non- responders. In addition, they reported that the fecal microbiome and metabolome, might impact the OC tumor's immune environment, affecting the response to ICB.

- The manuscript is well written, figures are self-explanatory and supplementary data is highly supportive of the observations. However, a few of my concerns are listed below:
-

Comments:

1. Introduction- Line 63-64: Reference not formatted

Response: We have now addressed this, and it has been corrected. Additionally, references have been updated accordingly.

2. In supplementary Fig1a, authors have observed that BRCA-positive patients had a 71.4% ORR compared to 30.4% in BRCA-negative patients ($p = 0.02$). What could be the possible reason behind this observation?

Response: Thank you for your comment. This study did not specifically investigate the difference between BRCA-positive and negative patients due to the limited number of BRCA+ patients enrolled in the trial.

- However, we were also intrigued by this finding and did additional analysis focused on the number of prior lines of therapy and the evaluation of peripheral blood samples from all 40 participants. We particularly assessed senescent and exhausted T cells in circulation. Our findings indicated that DCB to the triple regimen had a significantly lower number of circulating senescent T cells compared to LCB and factors such as age and BRCA mutation status also correlated with the degree of immune senescence, which aligns with expectations.

3. I understand that the authors have noticed abundance of some microbiome species in responders while some other in non-responders and the presence of several species was altered upon exposure to therapy. Firstly, is fecal sample the right biological specimen for studying differences in microbiome of ovarian cancer patients who respond to therapy versus non-responders. Secondly, how would you relate these findings in terms of biological cause and further implementation in treatment intervention?

Response:

Thank you for your insightful comment and questions.

Fecal Samples for Microbiome Analysis:

- During our study, we collected serial stool, skin, and vaginal microbiome samples, in addition to core biopsies of tumor tissue. While tumor tissue biopsies provide direct insights into the tumor microenvironment, their limited size and invasive nature make them less practical for longitudinal microbiome studies. Fecal samples, on the other hand, are non-invasive and offer a comprehensive view of the gut microbiome, which is the most densely populated microbial community in the human body. The gut microbiome has been

extensively studied for its significant role in modulating the immune system and influencing the efficacy of immunotherapies, making fecal samples a suitable and practical choice for our analysis.

- Although we collected vaginal microbiome samples, all patients had undergone hysterectomy, leaving only a blind vaginal pouch with no connection to the peritoneal cavity. Due to this anatomical consideration, and the lower relevance of the vaginal microbiome in influencing systemic immune responses compared to the gut microbiome, we prioritized our resources on analyzing fecal samples.
- Skin microbiome samples were also collected to potentially correlate with immune checkpoint inhibitor (IO) related skin toxicities. However, given that fewer than 5% of our patients experienced significant skin toxicity, the analysis of these samples was not pursued further.

Biological Cause and Implementation in Treatment Intervention:

- Our findings revealed distinct microbial compositions associated with patients who received DCB and LCB to the combination immunotherapy. Specifically, certain species such as *Intestinimonas butyriciproducens* and *Anaerotignum propionicum* were more abundant in DCB patients, while species like *Enterococcus faecalis* and *Clostridium perfringens* were prevalent in LCB patients. These microbial profiles suggest that the gut microbiome could be influencing the tumor immune microenvironment and the overall response to therapy.
- For instance, DCB patients exhibited elevated levels of butyrate-producing bacteria, which are known to enhance anti-tumor immunity by modulating T-cell function and reducing inflammation. In contrast, bacteria associated with LCB patients have been linked to resistance to immunotherapy. This microbial composition may impact the efficacy of immunotherapy by influencing systemic and local immune responses through the production of various metabolites, such as short-chain fatty acids, which are critical in immune modulation.

Further Implementation in Treatment Interventions:

- Given the high CD40 signaling observed in patients with DCB, we have initiated a new clinical trial, NCT05231122 where the corresponding author is the PI. This trial is currently funded by the Department of Defense Ovarian Cancer Research Program, in addition to Merck MISP funding, both awarded to PI. This trial investigates the combination of Pembrolizumab, Bevacizumab, and CDX-1140, a CD40 agonist antibody, in a randomized phase 2 clinical setting. The trial is currently enrolling patients at our institution and will expand to another major cancer center in a few weeks.
- In addition to this, our center is actively working on several other potential interventions as part of ongoing and upcoming clinical trials:

1. **Probiotic Supplementation:** We are exploring the introduction of beneficial microbes, particularly those producing butyrate, to potentially enhance the immune response in non-responders. By tailoring probiotics to enrich the gut microbiome with favorable bacterial species, we aim to improve patient outcomes.
 2. **Dietary Interventions:** We are investigating dietary modifications that promote the growth of beneficial microbiota. Diets high in fiber, for instance, support butyrate-producing bacteria, which could enhance the effectiveness of immunotherapy in ovarian cancer patients.
- Our goal through these clinical trials is to collect additional data on ovarian cancer patients, enabling us to better tailor immunotherapy approaches in the future. By leveraging these insights, we aim to develop more personalized and effective treatment strategies that enhance the efficacy of immunotherapies through microbiome modulation and other innovative approaches.
4. As this finding has a translational value, what would be future work in this direction?

Response: Please see the answer to the question above.